# Characteristics, Drivers, and Development Modes of Rural Space Commercialization under Different Altitude Gradients: The Case of the Mountain City of Chongqing

Zhenyi Lv [1,2], Aoxi Yang [1,2] and Yahui Wang [1,2,3,*]

1 School of Geographical Sciences, Southwest University, Chongqing 400715, China; lzy111@email.swu.edu.cn (Z.L.); xixi0201@email.swu.edu.cn (A.Y.)
2 New Liberal Arts Laboratory for Sustainable Development of Rural Western China, Chongqing 400715, China
3 Chongqing Jinfo Mountain Karst Ecosystem National Observation and Research Station, School of Geographical Sciences, Southwest University, Chongqing 400715, China
* Correspondence: wangyh1210@swu.edu.cn; Tel.: +86-158-0119-2532

**Abstract:** The spatial commercialization of rural areas is essential to achieve rural reconstruction and promote overall rural rejuvenation. Through the use of a land use transfer matrix and kernel density, this study uncovers the pattern characteristics, driving forces, and development patterns of rural spatial commodification at various altitudes, providing a scientific reference for rural spatial usage at various altitudes. The main conclusions of this study are as follows: (1) rural spatial commercialization is the result of land use transformation, and the differences in rural spatial commercialization development patterns lead to different characteristics in the local land use changes; (2) the implementation of urbanization, industrialization, and rural revitalization strategies has promoted the development of rural spatial commercialization to some extent; (3) There are significant differences in the characteristics of the land use change and the development pattern of rural space commercialization at various altitudes. The areas below 500 m are mainly for recreational projects that have a repeatable consumption and that are distributed in a concentrated and continuous manner. For such areas, the agglomeration effect should be taken full advantage of, and thus they should be developed in groups. The areas between 500 and 1000 m serve mainly the surrounding residents; the mode is based on the leisure and recreational projects with a block-shaped spatial distribution. In such areas, branded rural spaces with special features should be created. The areas above 1000 m are used primarily to construct tourist attractions and are dispersed in a point pattern. In such areas, the transportation conditions should be improved and the rural resources revitalized by designing reasonable travel routes.

**Keywords:** rural space commercialization; land use change; drivers; different altitudes; rural revitalization; China





## 1. Introduction

Due to urbanization and the mass exodus of rural labor, rural areas are facing problems such as the abandonment of arable land and the "hollowing out" of the countryside [1,2]. According to the 2022 Central Rural Work Conference, "Promoting rural revitalization comprehensively is an important task in building a strong agricultural country in the new era, with industrial revitalization being the most important task in rural revitalization. We should implement industrial support policies, rely on the unique resources of agriculture and rural areas, seek benefits from developing multiple agricultural functions and tapping the diverse values of the countryside, seek benefits from integrated development of one, two, and three industries, and improve market competitiveness and capacity for sustainable development". The commercialization of rural space is an important means of realizing the multiple values of the countryside and integrating industrial development [3]. The

overarching goal is to shift the mode of rural economic development from agricultural production to comprehensive business development, and from rough development to refined development, to achieve higher incomes for farmers, increased efficiency in agriculture, and prosperity for the countryside [4]. However, the commercialization of rural space in China is still in its infancy, and the methods and approaches are not yet mature enough to play a true role or have real value in rural revitalization. At the same time, the development mode of rural space commercialization varies by region. Therefore, to achieve the goal of rural revitalization, it is critical to explore the characteristics of the pattern, driving factors, and development models of rural space commercialization in various regions.

The definition of "space" has always been complex and multidimensional, and can be divided into material and immaterial space [5]. The book "The Production of Space" pioneered a "theory of spatial production" in which "space" could be continuously produced and re-produced through the practices of production and consumption [6]. The identification of the concept of rural space has always been a popular issue in rural studies [7,8]. Rural space is usually regarded as a synthesis of the physical, social, cultural, ecological, and other elements [9]. In this paper, rural space is defined as the space of rural production, with agricultural production at its core; the space of rural consumption, with rural tourism at its core; the space of rural residence, which includes rural settlements and rural architecture; and the space of rural landscape, with rural culture at its core [10–12]. According to Marx, commodities are the material embodiments of use-value, exchange value, and value, and commodification refers to the process of transforming or mutating things that are not originally part of the sale or circulation of goods, or acts that should not have commercial purposes, into objects that can be bought and sold under the conditions of the market economy [13]. The commercialization of rural space essentially refers to the dynamic process by which rural space acquires the characteristics and value of a commodity [14] and is "sold" for a profit under the conditions of a market economy [15,16]. Developed countries, such as those of Western Europe, the United States [11], and Japan [17], conducted earlier research on the commercialization of rural space [15,18], with a particular emphasis on the investigation of the mechanisms of its occurrence [19]. Furthermore, different stages of socioeconomic development have given rise to different theories of rural space development [20,21]. Following World War II, "productionism" emerged, which was characterized by a view of the countryside as a space for material production, with the production itself regarded as the ultimate measure of value and meaning [22]. In the 1990s, there was a shift away from an overemphasis on land resource production and toward a greater emphasis on diverse land-based economic activities. The countryside was viewed as a territorial spatial system with multiple values, combining material and immaterial production [8,23]. In addition, the theory of the transformation of the multifunctional countryside, which has been used to improve the relationship between the two, considers the countryside as having a variety of functions, including environmental management, ecological conservation, and cultural heritage [24,25]. Simultaneously, foreign scholars have conducted a large number of case studies on the phenomenon of the commercialization of rural space, investigating how to create new tourism resources through the commercialization of rural space to achieve regional development [26].

Chinese scholars' research on the commercialization of rural space is still at an early stage, and their research focuses primarily on the combination of the foreign theories on the commercialization of rural space, the inspiration of typical foreign cases in China, and the empirical research on the reconstruction of rural areas through the commercialization of rural space in developed regions. The research areas are concentrated primarily in the developed plain regions such as Beijing, Jiangsu, and Guangdong; the research methods mainly include the actor-network theory [27], the case study method [28], and the spatial analysis methods, such as kernel density estimation [29,30]. Due to China's rapid urbanization, the vast Chinese countryside has gradually transformed from a single production function to a multi-functional integration of production, living, ecology, aesthetics, and education, and the implicit value of rural space has gradually emerged [31,32]. Since

rural development is relatively weak in China's mountainous regions, which make up 69% of the nation's total area, it is especially crucial to realize the commercialization of rural space in these areas to facilitate the realization of rural revitalization. However, the existing research on the commercialization of rural space is focused on the plain regions, and there are fewer studies on the phenomenon of spatial commercialization in the vast mountainous countryside. Moreover, there is a lack of systematic studies on specific cases in typical regions. In studies on the commercialization of rural space in mountainous areas, the current classification method is still based on that of the plains, which is based on the functional zoning of cities or the spatial circle from the inside out, ignoring the unique influence of altitude on rural development in mountainous areas. Due to this, it is challenging to use rural spatial commercialization in mountainous regions as a driving force behind the overall revitalization of the countryside.

In view of this and in the context of the accelerating urbanization and the rising demand for non-farm livelihoods by farming households [28], this paper selects Chongqing as a typical representative region within the mountainous region; takes the spatial commercialization of the countryside as the entry point; uses the land use data and the data on the types of spatial commercialization of the countryside as support; employs the land use change measurement and kernel density estimation methods to clarify the characteristics of spatial commercialization of the countryside at different altitudes; analyses the driving factors behind the formation and development of the spatial commercialization of the countryside; and reveals the problems in the current development model. The aims of this paper are to help raise the income of farming households and to help achieve sustainable economic and social development in the countryside. At the same time, through comparative studies, the development patterns of the commercialization of rural space at different altitudes are considered, the rational and effective use of rural space is promoted, and the path of coordinated regional development is explored.

## 2. Materials and Methods

### 2.1. Study Area

Chongqing is in southwest China, near the upper reaches of the Yangtze River, with western Hubei to the east, Sichuan-Guizhou to the southwest, and Sichuan-Shanxi to the northwest. The following are the reasons for selecting it as the topic of this paper: Firstly, the region's mountainous and hilly landscape accounts for 70% of the land area, with an altitude difference of 2723 m, and significant geographical variations in natural resource endowments, which can effectively represent the development of rural spatial commodification at different altitudes. Secondly, by the end of 2021, the city's resident population reached 32,124,300 people, including 9,533,000 rural residents, the per capita disposable income of whom was RMB 18,100, representing only 41.6% of the per capita disposable income of the urban residents; these factors demonstrate the typical geographical background of a "big city, big rural area, and big mountainous area" [30]. Thirdly, progress has been made in the commercialization of the rural space in Chongqing, as demonstrated by the following developments: the development of clusters of advantageous rural industries with special characteristics, the innovative development of rural tourism, and the emergence of new industries and new business models. Due to the large size of the study area and the complex and diverse topography, this paper selects Xiema Town in the Beibei District, Xinglong Town in the Yubei District, and Xianushan Town in the Wulong District as the typical representative areas in three types of altitude intervals: below 500 m, between 500 m and 1000 m, and above 1000 m, all within one to two hours' drive to Chongqing's main city (Figure 1).

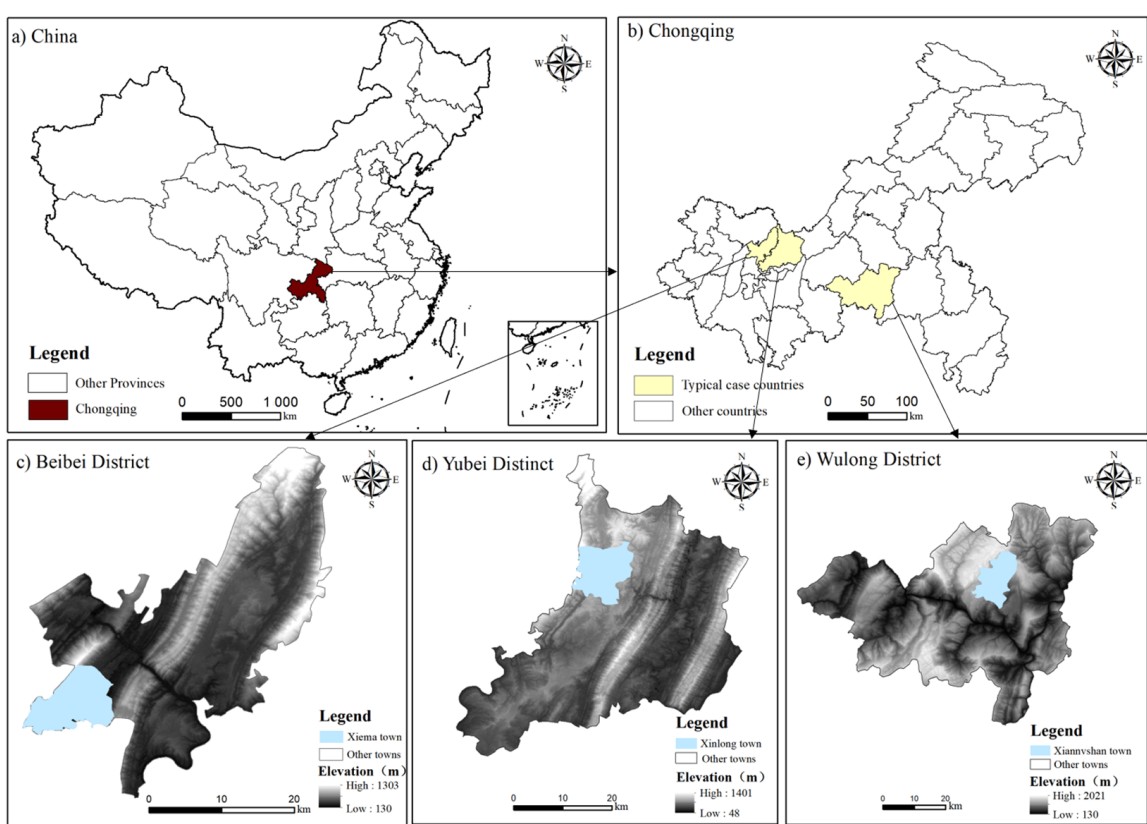

**Figure 1.** Schematic diagram of the study area.

The Beibei District, one of Chongqing's main urban areas, is a typical case area in an area below 500 m, with unique location conditions and the spatial advantage of integrated scenic and urban areas. Xiema Town, located in the southwest of the Beibei District, is an area of 58 square kilometers and flat and open terrain. The Yubei District, located in the northwest of Chongqing, is a typical case area in the area between 500 m and 1000 m, and belongs to the main urban area of Chongqing and the metropolitan area of Chongqing, and the urban and rural development in the region is very different. Xinglong Town, located in the north-central part of the Yubei District, is one of Chongqing's ecological towns, with the advantage of "facing the city in front and the countryside behind", covering an area of 93.5 square kilometers. The Wulong District, located southeast of Chongqing in the lower reaches of the Wujiang River, the Wuling Mountains, and the Great Lou Mountain Gorge, is a typical case area above 1000 m, and includes the World Natural Heritage Karst Furong Cave and the national 5A tourist attraction of the Tiansheng Three Bridges. Xiannushan Town, located in the north-central part of the Wulong District, Chongqing, as well as 20 km from the Wulong City District, is an important location for economic growth in the Wulong District's "one river, two wings" development strategy. Because of its resource advantages, its rural space commercialization has a higher degree of development than other areas at the same altitude, which can serve as a model for other regions.

*2.2. Data*

The required data for this article mainly consist of regional land use data and rural spatial commercialization representation data. The land use data used in this article include topographic maps of Chongqing, the current land use map of the study area, and other supporting maps from Wuhan University's research paper "30 m annual land cover and its dynamics in China from 1990 to 2019"; the study area's socioeconomic development data were obtained from the sixth and seventh population censuses, the statistical yearbooks published by the Chongqing Municipal Bureau of Statistics, and the statistical bureaus of the

districts and counties; the data on the types of spatial commercialization of the countryside mainly consisted of four types: hotels and lodges, recreation and entertainment, such as farmhouses and picking gardens, scientific and educational culture, such as museums, and tourist attractions. The data are obtained by first using Python to obtain the names, categories, administrative regions, latitude, and longitude of the relevant geographical elements in the Gaode map, and then obtaining the corresponding POI data of rural space commercialization in Chongqing.

*2.3. Methods*

2.3.1. Estimation of Land Use Change

The total change in land use type is the sum of the transfers in and out of a specific land use type, as well as the sum of a specific land type's net change and exchange change [33]. The following is the formula for this:

$$S_i = S_{gain} + S_{loss} = D_i + C_i \tag{1}$$

$$S = S_{gain} - S_{loss} \tag{2}$$

where $S_i$ represents the total change of land-use type i, $S_{gain}$ is the transfer in of land-use i, and $S_{loss}$ is the transfer out of land-use i. $D_i$ is the net change of land-use i, where "$\pm$" of the $D_i$ value indicates the direction of change of land-use i, "+" indicates a net increase, and "$-$" indicates a net decrease. $C_i$ is the amount of exchange change of land-use i.

The net change in land use is the absolute change in land use and is one of the common indicators of land use change. However, due to the fixed and unique spatial location of land use, the net change volume has limitations and cannot objectively and accurately reflect the dynamic process of the interchange of land classes. Therefore, the amount of land exchange change should be used as one of the measurement indicators that can quantitatively analyze the dynamic amount of change in the interconversion of one land class with another. The greater the degree of change in a land class, the greater the amount of exchange change; conversely, the smaller the degree of change in a land class, the smaller the amount of exchange change.

2.3.2. Measure of Land Use Change

The magnitude of land use change refers to the amount of change in land use type relative to the total area of the study area over a fixed period of time, which can be used to analyze the overall trend for land use change and can directly characterize the rate of land use change. The formula is as follows:

$$U = [(S_b - S_a) \div S \times 100] \times 100\% \tag{3}$$

where U is the magnitude of land use change for a land category, $S_a$ is the area of a land category at the beginning of the study period, $S_b$ is the area of a land category at the end of the study period, and S is the total area of the study area. The "$\pm$" value of U indicates the direction of change of land-use i, "+" indicates a net increase, and "$-$" indicates a net decrease.

Land use change dynamics refers to the rate at which land use types change in quantity over a fixed period, and it can be used to forecast future trends in land use change. The formula is as follows:

$$V = [(S_b - S_a) \div S_a] \div T \times 100\% \tag{4}$$

where V is the dynamic attitude of the change in land use for a particular land category. T For the duration of the study, the same interval of 10 years was used in this study, T = 10. Where the value of "$\pm$" indicates the direction of change of land-use i, "+" indicates the net increase, and "$-$" indicates the net decrease.

### 2.3.3. POI-Based Kernel Density Estimation

POI data describe a geographic entity's spatial and attribute information, such as its name, category, and coordinates [34]. Kernel density estimation, proposed by Emanuel, is a geographic algorithm for calculating the density of point features or line features. The principle is that the closer the thing is to the core feature, the greater the density expansion value; it is frequently used to study the spatial distribution characteristics of a group of points [35]. As of March 2022, the number of major rural space commoditization such as rural B&Bs and farmhouses in Chongqing is 22,409. This paper investigates the degree of agglomeration of rural space commercialization in Chongqing based on the distribution of the kernel density values using the POI data of country houses and farmhouses in Chongqing. The search radius is set to 10,000 m using the kernel density estimation of the POI data obtained, and the output image element size is set to 40 m. The formula is as follows:

$$f(x) = \frac{1}{nh} \sum_{i=1}^{n} K \frac{x - x_i}{h} \tag{5}$$

where $f(x)$ is the kernel density function at the spatial location $x$; $h$ is the analysis range threshold, i.e., the search radius; $K$ is the default spatial weight kernel function; and $x - x_i$ represents the distance from the valuation point to the output grid.

## 3. Results

### 3.1. Characteristics of Land Use Change in the Case Area

The typical case areas of the three elevation intervals, below 500 m, 500–1000 m, and above 1000 m, are Xiema Town in the Beibei District, Xinglong Town in the Yubei District, and Xiannushan Town in the Wulong District, respectively. They present different land use characteristics (Figure 2 and Table 1). The development status of the commodification of rural space varies, as does the impact on the pattern of land use change [36].

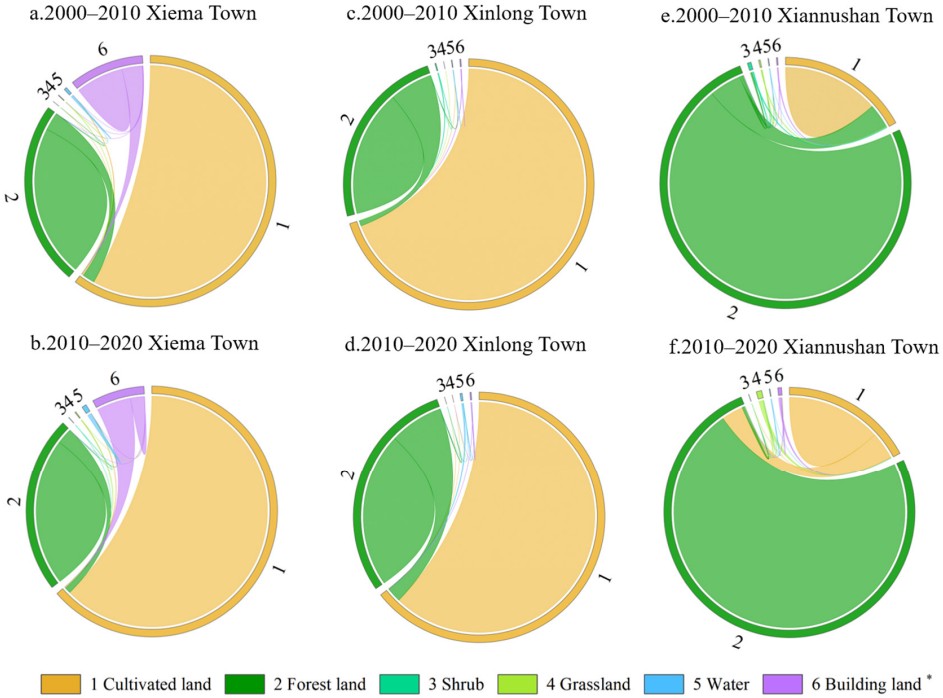

**Figure 2.** Land use change chord map of typical case areas from 2000 to 2020. * The trajectory lines of different colors denote the flow direction of a particular place in a specific period. The thickness of the trajectory lines denotes the variation; the larger the variation, the thicker the trajectory lines.

**Table 1.** Statistics on the amount and characteristics of land use change in typical case areas from 2000 to 2020.

| Land Use Type | Total Change(km$^2$) | | | Swap Change(km$^2$) | | | Net Change(km$^2$) | | | Change Amplitude (%) | | |
|---|---|---|---|---|---|---|---|---|---|---|---|---|
| | Xiema Town | Xinlong Town | Xiannushan Town | Xiema Town | Xinlong Town | Xiannushan Town | Xiema Town | Xinlong Town | Xiannushan Town | Xiema Town | Xinlong Town | Xiannushan Town |
| Cultivated land | 8.51 | 13.24 | 15.92 | 14.85 | 23.90 | 17.53 | −6.33 | −10.66 | −1.61 | −0.69 | −0.73 | −0.33 |
| Woodland | 3.15 | 12.95 | 16.13 | 1.67 | 2.53 | 14.94 | 1.48 | 10.42 | 1.20 | 0.52 | 2.94 | 0.06 |
| Shrub | 0.00 | 0.01 | 0.65 | 0.00 | 0.00 | 1.24 | 0.00 | 0.01 | −0.60 | −5.00 | | −4.75 |
| Grassland | 0.04 | 0.00 | 1.38 | 0.00 | 0.00 | 1.01 | 0.04 | 0.00 | 0.37 | | | 3.33 |
| Water | 0.32 | 0.26 | 0.02 | 0.54 | 0.10 | 0.02 | −0.23 | 0.16 | 0.00 | −2.46 | 7.79 | 0.15 |
| Building land | 5.12 | 0.09 | 0.64 | 0.08 | 0.01 | 0.00 | 5.04 | 0.08 | 0.64 | 19.30 | 5.01 | 926.93 |

First, overall, the typical case areas at all elevations demonstrate mainly a decrease in arable land and an increase in forest land and construction land. Arable land is the land type with the greatest change in terms of magnitude, and forest land and construction land contribute the most to both the transfer-in and transfer-out structures. The exchange of arable land for forest land accounts for most of the change in land use type. Because the background value of rural construction land is low, the change dynamics of construction land are all higher. In particular, the town of Xianushan in the Wulong District is located in a mountainous area above 1000 m, with a small proportion of construction land to the total land area. Social capital has been aggressively introduced for the construction of scenic spots since the commercialization of rural space, and the dynamics of change in the construction land during the study period reached 926.93%, with a much higher rate of change than the other land use types.

Secondly, in comparison, in terms of the number of changes in each category, forest land is the land use type with the greatest increase in area in Xinlong Town, Yubei District, and Xiannushan Town, Wulong District, while construction land is the land type with the greatest increase in area in Xiema Town, Beibei District. This is mainly because Xiema Town in the Beibei District is located in a low-altitude area, below 500 m, where rural commercialization is rapidly developing, concentrated in patches, and has a broad reach. Thus, the modern urban agricultural gardens and farm caravans have a large area of land for construction and are rapidly growing. In terms of the magnitude of change in each category, the magnitude of change in the water bodies is small in Xiema Town of the Beibei District and Xiannushan Town of the Wulong District. Still, it is second only to that of cultivated land and forest land in Xinglong Town of the Yubei District, where the attitude of change in the water bodies is the greatest. This is primarily because Xinglong Town in the Yubei District has constructed an ecological area covering approximately 3000 acres as part of the spatial commercialization process, resulting in a rapid expansion of the water body area.

In summary, the commercialization of rural space is essentially the result of a countryside reconfiguration, which has resulted in an increase in the function of spatial consumption, the root cause of which is land use transformation. The improvement of the land use function is an important way to implement the rural revitalization strategy [37]. By analyzing the characteristics of land use change in different altitude case areas, differences in the development mode of rural spatial commercialization which result in different local land use change characteristics can be found. Therefore, choosing a suitable spatial commercialization development mode for the region can form a rational, efficient, and intensive land use structure by changing the regional land use, and ultimately maximize the comprehensive economic, social, and ecological benefits of land use [38]. At the same time, the original land use structure of the region can, to some extent, reflect the transformation of the regional land functions in a certain period [39] and provide the right direction for the selection of the spatial commercialization development mode.

### 3.2. Characteristics of Rural Space Commercialization in the Case Areas

Figure 3 shows the distribution of rural space commercialization types in typical case areas. Figure 4 shows the rural space commercialization nuclear density map in typical case areas.

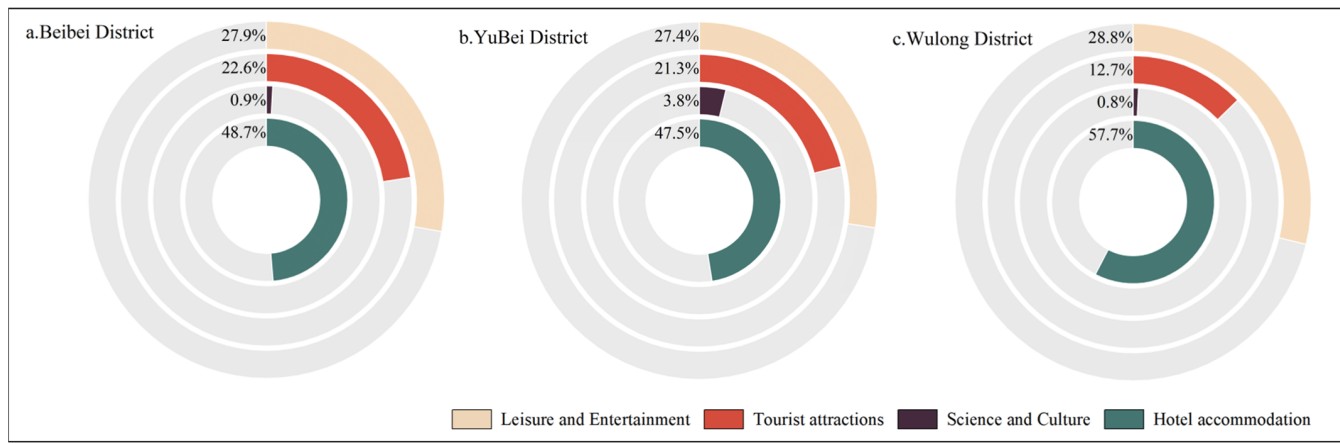

**Figure 3.** Distribution of rural space commercialization types in typical case areas.

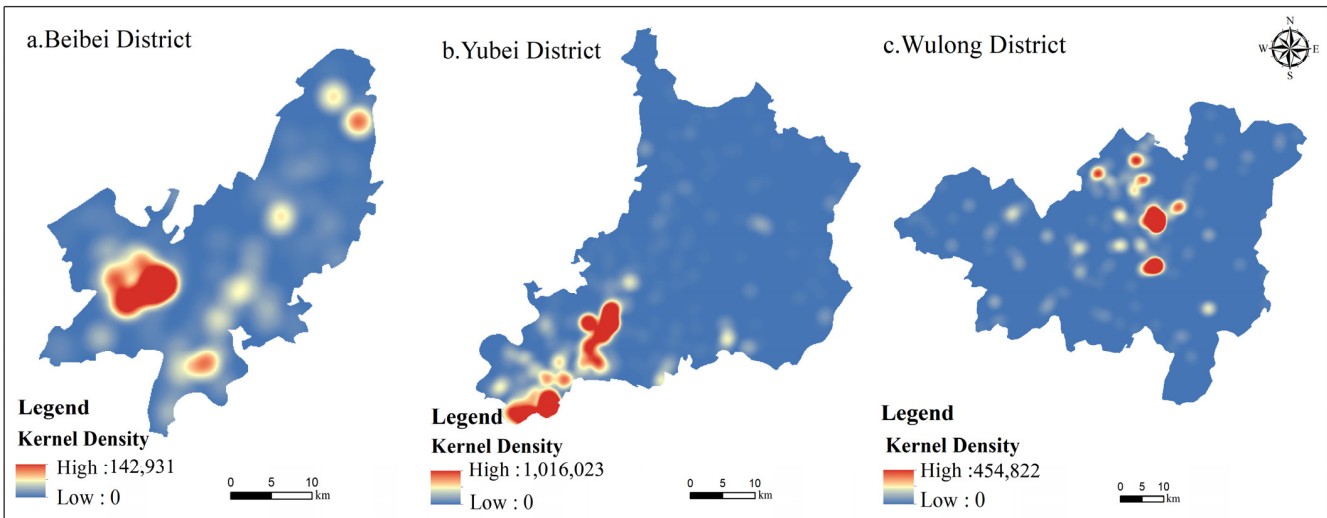

**Figure 4.** Rural space commercialization nuclear density map in typical case areas.

The Beibei District, located in a low-altitude area below 500 m, has a high concentration of rural space commercialization, and its overall spatial layout is distributed in a faceted manner. Further, its rural space commercialization is dominated by leisure and entertainment venues such as farm caravans and picking gardens. In 2009, Xiema Town in the Beibei District aggressively developed new rural industries and actively guided the development of upscale farmhouse catering services. Overall, this stage of the rural space is still dominated by the production functions, while the rural space consumption function is being gradually developed. In 2018, the comprehensive agricultural leisure demonstration park Jiang Zhou cherry garden opened to the public, attracting visitors from the main city and surrounding areas. In 2019, Chongqing Cloud Agricultural Development Co., Ltd. from Chongqing, China, was introduced to create a collection of leisure, tourism, picking, and holidays in one of the modern urban agricultural parks, and eventually achieved an increase in collective income of 8000 yuan. The Beibei District, Xiema Town, integrated regional resource characteristics in each countryside to create agritourism or special crop picking gardens and other recreational and entertainment projects that can be consumed

repeatedly by villagers' self-owned or corporatized rural resource developments. This attracts visitors from the surrounding and main urban areas. Space commercialization characteristics are obvious.

The Yubei District, located in a mid-latitude area of between 500 m and 1000 m, has a block-shaped spatial distribution of the spatial layout of rural space commercialization, including recreational and tourism attractions, two types of rural space commercialization, developed together. In 2004, the town of Xinglong in the Yubei District introduced capital to develop local special agriculture, establishing a foundation for plum cultivation. In 2010, the town held a plum culture festival, shifting from production to the development of rural spaces through the creation of tourism brands. In 2017, Xinglong Town adopted the agricultural park as its core leader and focused on assisting in the construction of projects such as the Xinglong Flower Sea and the outdoor camping base to promote the development of rural commercialization through the integration of multiple industries. In 2020, the town fully exploited the location's advantages, encouraged the deep integration of "agriculture, culture, tourism, and creativity," and effectively tapped into tourism resources, thus promoting the process of the commodification of rural space. In general, the commercialization of the rural spaces in Xinglong Town, Yubei District is slow, and the development potential remains untapped.

The Wulong District, located at a high altitude of above 1000 m, has a dotted pattern of rural spatial commercialization, with a focus on local characteristic alpine resources. Its development of spatial commercialization of the tourist attraction type is remarkable. The town of Xiannushan in the Wulong District emerged earlier than the surrounding areas in the commodification of rural space, becoming a model area for the development of rural space commodification in the region thanks to its unique natural resources. In 2002, the Furong River National Key Scenic Spot in the Wulong District was approved and completed, and the Harbor Peninsula became a popular tourist attraction. In 2012, the Chongqing Intangible Cultural Heritage Base was built. In 2021, the government actively promoted key projects in Xiannushan District, constructing a demonstration base for youth science in Xiannushan District, a town with cultural and artistic characteristics as well as an international ecological recreation town, and fully utilized the "Internet +" and other emerging means to promote the two-way extension of the rural industry chain and drive the sustainable development of rural space commercialization. Overall, the Wulong District is a relatively unique area for the commercialization of rural space above 1000 m, with a much higher degree of development than other areas at the same altitude and a focus on creating tourist attractions.

*3.3. Driving Mechanisms of Rural Space Commoditization in the Case Areas*

In recent years, the phenomenon of the commercialization of rural space in China has arisen and grown along with the ongoing transformation of the countryside [40]. Earlier studies explicitly attributed the "external aid drive" as the cause of the commodification of rural space, but most scholars today think that the cause should be an "internal and external combined drive" [41]. Since the reform and opening, the rapid development of urbanization and industrialization in China has led to a drastic transformation and reconstruction of rural areas. It has produced specific responses to rural space [23]. Urban capital has flowed into the countryside in large quantities; the market economy has prompted changes in the function of rural land; the traditional production function has been weakened; the consumption function has been continuously developed; the land has continuously increased in value [42]; and the rural industrial structure has gradually shifted from primary industries to secondary and tertiary industries [5,43]. This has promoted the conversion of rural areas from traditional production to contemporary consumption spaces, as well as the formation and growth of the commercialization of rural space in China. The connection between the rural and urban areas in the region is distinct due to the varying location conditions at various altitudes, which also encourages the development of various rural space commercialization characteristics and development paradigms(Figure 5).

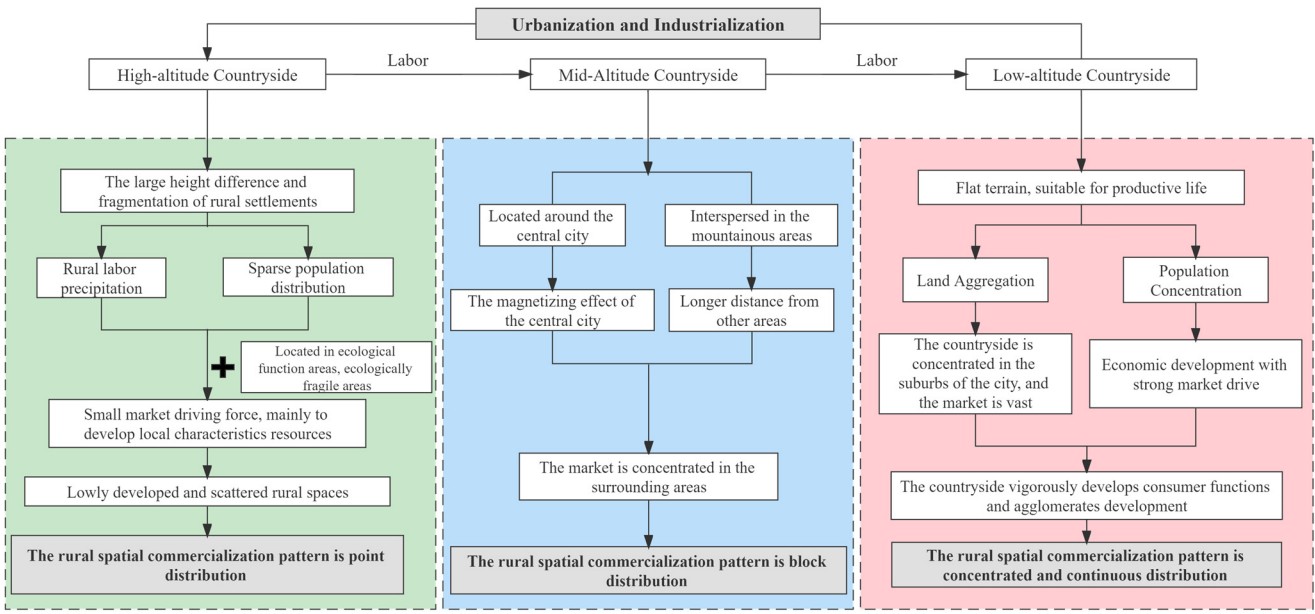

**Figure 5.** Driving mechanisms of spatial commoditization at different altitudes.

The rural spatial commercialization pattern in low-altitude areas below 500 m is concentrated and contiguous. Such areas in Chongqing are concentrated primarily in the central urban areas, such as the Yuzhong District, the Jiangbei District, and the Nanan District, where the terrain is flat and suitable for human habitation and development, the economy is developed, and the population is concentrated. As a result, the people who benefit from the region's commercialization of rural space are primarily urban residents in central urban areas, where there is a high demand for quality rural tourism. At the same time, the villages in the central urban areas are distributed throughout the suburban areas, with good economic foundations, an early start of habitat improvement, and near-perfect public service facilities, all of which provide a good material foundation and spatial environment for the commercialization of the local rural space.

The spatial commercialization of the rural areas between 500 and 1000 m in elevation exhibits a blocky distribution pattern with a higher density. The area is mostly concentrated around the central urban area or is interspersed with mountainous areas, such as in Xinglong Town in the Yubei District. On the one hand, the rural economy in this altitude range is more backward in comparison to the development in the central urban area. A large number of people have left the area due to the magnetic attraction effect of the core urban region, and therefore the market for the commercialization of the rural territory has dwindled. On the other hand, compared to the areas below 500 m, the villages in areas between 500 and 1000 m in elevation are more dispersed and affected by the influence of driving. Therefore, the market demand for the commercialization of the rural space comes primarily from the residents of the surrounding cities, making it less appealing to the cross-area population.

The rural spatial commodities are dotted, sparse, and scattered in the high-altitude areas above 1000 m. Such areas, represented by Xiannushan Town of the Wulong District, are primarily distributed in the northeastern and southeastern regions of Chongqing, having special geographical locations. Northeast Chongqing is part of the Three Gorges Reservoir Area, which serves as an important ecological barrier in the Yangtze River's upper reaches; southeast Chongqing is a national key ecological function area, an important biodiversity reserve, and an ecological folk culture tourism belt [44]. The villages in this altitude range are both ecologically sensitive and ecologically fragile, and development is based primarily on the "protection on the surface and development on the point" approach. As a result, such locations have experienced some degree of ecological migration, are economically disadvantaged, and are underdeveloped, with a sparse population distribution. At the same

time, due to the large difference in terrain height and relatively low accessibility, cross-zone comprehensive development is difficult, and the development of local characteristics is mainly independent.

## 4. Discussions

### 4.1. Overall Pattern of Rural Spatial Commercialization in Chongqing

The overall pattern characteristi0063s of rural space commercialization are obtained after analyzing the spatial distribution of rural space commercialization in Chongqing (Figure 6).

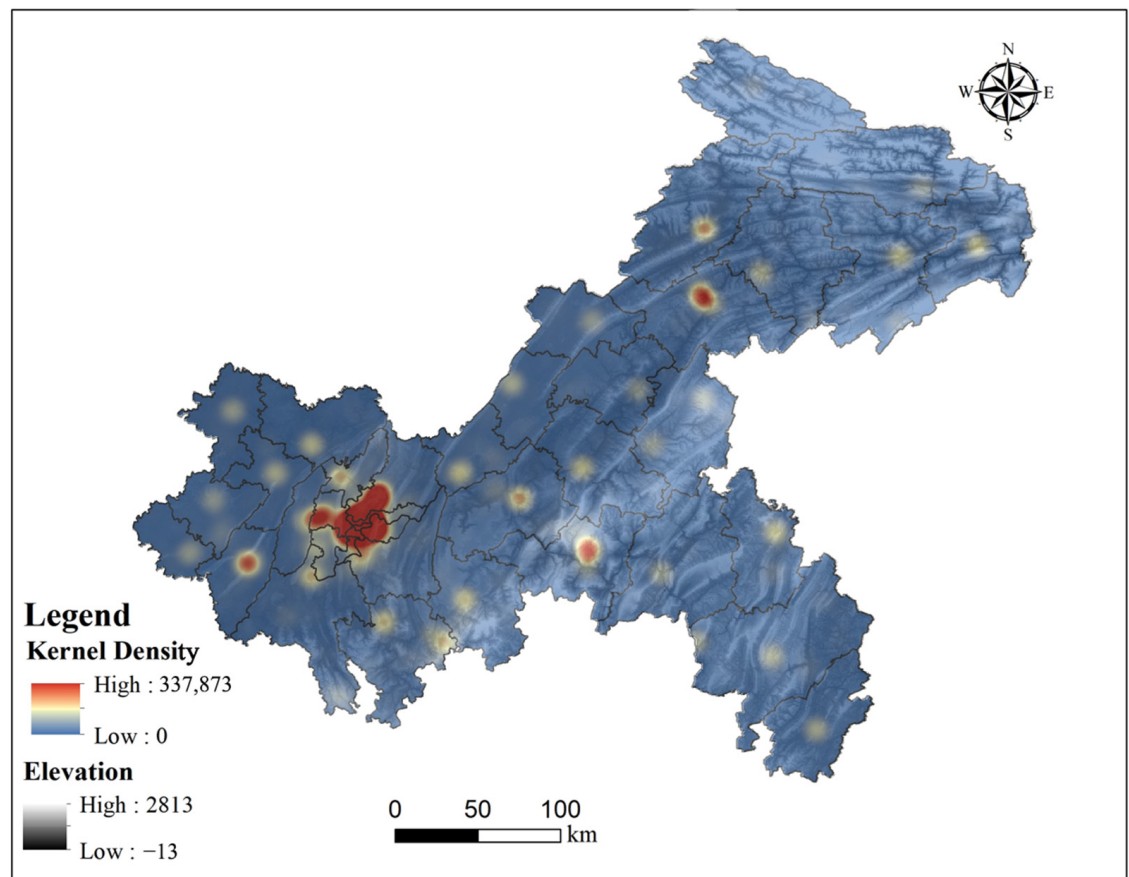

**Figure 6.** Nucleation density of rural space commercialization in Chongqing.

The commercialization of rural space in Chongqing forms a high-density ring in the main urban area, as well as three high-density zones in the Yongchuan, Wulong, and Wanzhou districts; the medium-density zones are distributed primarily in the hilly areas with small height differences; and low-density zones are scattered and widely distributed. The spatial commercialization of the countryside is concentrated mainly in the central urban area or the countryside around tourist attractions, such as Xiannushan in the Wulong District, based on the distribution characteristics of each density zone. Rural space commercialization is concentrated in the plain areas and sparsely distributed in mountainous areas. The development of rural space commercialization occurs when high-profile tourist and scenic spots exist in the high-altitude mountainous areas. Overall, the regional development of rural space commercialization in Chongqing is uneven.

### 4.2. Insights into the Commercialization of Rural Space in Different Altitudes

Low-altitude areas—those below 500 m—focus on integrating regional resources and achieving cluster development. Areas in this altitude range should integrate the characteris-

tics of the region's rural resources, take full advantage of the area's topographic advantages, and utilize the clustering effect. For example, when integrating the regional resources to organize the relevant festivals and events, these areas should set up special parks with different themes to achieve a mutual attraction between villages in the region with a series of activities, and thereby help the coordinated development of the commercialization of the rural space in the region using the differences in the agricultural production in different villages in the region. Concurrently, because the land in the region is mostly concentrated and contiguous, with superior agricultural planting conditions, local enterprises should be encouraged to improve the level of science and technology, improve the quality of agricultural products, develop various functional food and health products, develop superior seeds and products, enrich the product line of deep cultural experiences, and realize the transformation of the agricultural products into the tourism industry based on a strict adherence to the red line of arable land.

In middle-altitude areas—those between 500 m and 1000 m—due to the small regional market radiation range, the key to growing rural space commercialization is to build branded rural spaces, improve the rural visibility, and broaden the market radiation range. For example, because such areas typically lack obvious topographic and landscape characteristics, the process of space commercialization can create branded rural spaces through visual image shaping. Simultaneously, the breadth of the rural spatialized products should be investigated and actively promoted to the outside world. The link between the products and the cultural experience, aesthetic services, and artistic creativity should be strengthened, as should the integrated development of various functions, such as rural space idyllic tourism, leisure picking, cultural experiences, and science education.

In high-altitude areas—those above 1000 m—the priority is to increase the investment in road construction, improve the transportation infrastructure, enhance the traffic conditions, and improve accessibility and convenience. Such areas can also carry out the design of tourism routes mainly for the natural experience and sightseeing before the development of the area by transforming mountainous areas to their advantage, attracting tourists who enjoy driving and improving the attractiveness of the areas. Simultaneously, because villages between such altitude zones are dispersed and thus it is difficult to form independent attractions, the core characteristics of each village in the region can be explored deeply and linked together to form a theme promotion during the planning and development. Then, in different villages, the priority is to construct the same type and different series of facilities to deepen the tourists' memory of the association between attractions, so that the tourists' impression of the joint theme of the attractions is continuously deepened and the tourists' return rate is improved.

### 4.3. Implications of the Rural Space Commodification for Rural Revitalization

The rural revitalization strategy is a major strategic decision affecting China's agriculture and rural farmers; it was proposed in the 19th Party Congress report, which points out the direction for the national government to do a good job with the "three rural areas" in the current and future periods. In the new era, China's agricultural supply-side structural reform has made significant progress, the agricultural production capacity has been significantly increased, new industries and business models are flourishing in the rural areas, and profound changes are occurring in rural society. However, the rural areas remain the weakest link in China's modernization plan, with the most obvious shortcomings in economic and social development concentrated in the "three rural areas." The current situation of a poor rural foundation and lagging development remains a real issue that must be addressed. The 20th Central Document No. 1, "Opinions of the Central Committee of the Communist Party of China and the State Council on the Key Work of Promoting Rural Revitalization in 2023," emphasizes the importance of promoting the high-quality development of rural industries and broadening the channels through which farmers can increase their incomes. The prosperity of industry is the root of the rural revitalization strategy, and the commercialization of rural space is an important practice to revitalize

rural space resources and promote the transformation and upgrading of rural areas from traditional production spaces to modern consumption spaces.

The key to rural space commercialization is activating rural resources, and reproducing and recreating the value of rural spatial resources [45]. The first step toward activating rural space is to fully explore local resources, and the key way to realize the transformation of its multi-functional value is to explore the typical path of the commercialization of spatial resources on this basis. Because China's rural areas are vast and the basic conditions vary greatly from place to place, the scientific commercialization model must be based on each place's differentiated positioning. Whether we focus on agricultural production to create special agricultural products or extend the industrial chain, the goal is to promote the transformation of rural space from a single agricultural production space to a multifunctional space combining production and consumption. Concurrently, the process of commercialization of rural space is inextricably linked to the influence of consumer culture and industrial capital intervention, which will hasten the dramatic changes in the countryside and trigger its reconstruction. As a result, to effectively contribute to rural revitalization, we must also be aware of and avoid potential risks associated with the commercialization of rural space [1].

## 5. Conclusions

In the context of the rural revitalization strategy, the Chinese countryside urgently requires the removal of barriers to development and the resolution of the dilemmas of serious land abandonment, the single inefficient function, and the inadequate improvement of the human living environment. Rural space commercialization is an important means of rural reconstruction and spatial transformation, and it is critical to investigate its characteristics, driving factors, and modes for the implementation of a rural revitalization strategy. In recent years, Chongqing has been aggressively developing rural tourism around rural revitalization and poverty alleviation, as well as promoting the process of rural spatial commercialization. At the same time, Chongqing, a mountain city, has a unique geographical environment with an undulating terrain and strong three-dimensional qualities. Therefore, the development of rural space commercialization varies greatly and has distinct characteristics. This study examines the mechanisms that drive the spatial commercialization in rural Chongqing and discovers that urbanization, industrialization, and rural rejuvenation efforts are major driving forces for the development of spatial commercialization in rural areas. Initially, urbanization and industrialization facilitated a shift in the spatial pattern of rural land use, and the introduction of foreign capital prompted the expansion of rural areas from a single production function to a multi-functional integrated development, resulting in the emergence of rural spatial commodification. Later, the implementation of a rural revitalization strategy improved the material basis and spatial environment for the development of rural space, optimized the rural land use structure, and promoted the integration of rural multi-industries and the re-exploitation of rural space. These factors facilitated the development of rural space commercialization.

This paper uses Chongqing as an example and discusses the characteristics of rural space commercialization at different altitudes, putting forward strategies to optimize the development mode of rural space commercialization, release the vitality of rural space, promote the high-quality development of rural industry, promote the continuous increase in farmers' income, promote rural revitalization, and achieve common prosperity. In addition, in China, the commercialization of rural spaces is still in its early stage of development. The Chongqing Municipality is used as the research object in this paper, which lacks a comprehensive discussion on it. Further research needs to be carried out in combination with the actual situation of different regions in China to carry out more details discussions in theory and practice.

**Author Contributions:** The authors contributed together to the completion of this article. Specifically, their individual contributions are as follows: conceptualization, Y.W. and A.Y.; methodology, A.Y.; software, Z.L. and A.Y.; validation, Y.W. and Z.L.; formal analysis, Z.L.; investigation, Z.L.; resources,

A.Y.; data curation, Z.L.; writing—original draft preparation, Z.L.; writing—review and editing, Z.L.; visualization, Y.W.; supervision, Y.W.; project administration, Y.W.; funding acquisition, Y.W. All authors have read and agreed to the published version of the manuscript.

**Funding:** This research was funded by the National Training Program of Innovation and Entrepreneurship for Undergraduates (202210635022), the National Natural Science Foundation of China (42271263, 41901232), and the Fundamental Research Funds for the Central Universities (SWU-KT22008).

**Institutional Review Board Statement:** Not applicable.

**Informed Consent Statement:** Not applicable.

**Data Availability Statement:** Not applicable.

**Acknowledgments:** Thank you to everyone who contributed to this study.

**Conflicts of Interest:** The authors declare no conflict of interest.

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
