# Peer review of "Characteristics, Drivers, and Development Modes of Rural Space Commercialization under Different Altitude Gradients: The Case of the Mountain City of Chongqing"

_land, doi:10.3390/land12051028_

Round 1

Reviewer 1 Report

TITLE

The article’s title is suitable with the content of the paper, the title of the paper: Characteristics, drivers, and development modes of rural space commercialization under different altitude gradients: The case of the mountain city of Chongqing.

ABSTRACT

The abstract is well-designed and briefly express the present research thus being of interests and readable thus capturing the reader’s attention. It present in an appropriate manner the main research hypothesis, the problem statement, the methods and the main findings.

KEY WORDS

The key words are appropriate to the present research and are clearly stated.

ORIGINALITY

The article meets a high level of originality from my side argued by the main research theme and the research hypothesis. Furthermore, the originality of the paper is highlighted by the main results of the paper. The authors construct a well-designed theoretical background closely related to the current specialised literature in the field.

THE PAPER S STRUCTURE

The structure of the paper is correct in line with the journal standards and meet the publication requirements considering the paper logic. The objectives seem to be clear formulated as well as the investigation is drawn. The core argument of the paper illustrates the paper relevance and the research originality. Introduction part, literature review is weak, it should definitely be improved. The results are clearly express and well connected both to the theoretical framework and discussions.

 THE METHODS

The methodological design is appropriate and the methods fit well to the present investigation. But, the methodology can be explained in a little more detail.

THE MAIN ANALYSIS

The main research is well design and appropriate conducted in line with the main questions in spatial planning in the investigated areas.

CONCLUSIONS

The conclusions fit well summarising the main ideas of the present analysis but I would expand a little bit this section.

 THE ENGLISH LANGUAGE

I think the English is ok as far as I could see but I am not a native speaker.

Reviewer 2 Report

I found your research very interesting. I really appreciate this geographical approach of your research and GIS analytics.

At the begging, reconsider the title of your paper since you`re researching Chongqing city but secondary mountain urban centers in this city region. So maybe “… The case of mountain urban centers of Chongqing city region” title should be more appropriate.

Regarding analysis of rural areas my opinion is that some comments about demographic situation and potential for development is missing. It is well known that even the best ideas and plans will not work if population/community is not prepare to complete proposed tasks. This is even more represented in the rural areas, where usually older, undereducated, poorer population is settled. Furthermore, if such rural area is dominated by much more developed urban center then will be hard to stop immigration and abandonment of rural space. Thus I`m strongly suggesting to you to add one section about demographic and socio-economic issues of these rural areas to your research.

Although text is clear and relatively simple I`m suggesting proof reading by English native speaker.
